# Unblocking Fine-Grained Evaluation of Detailed Captions: An Explaining AutoRater and Critic-and-Revise Pipeline

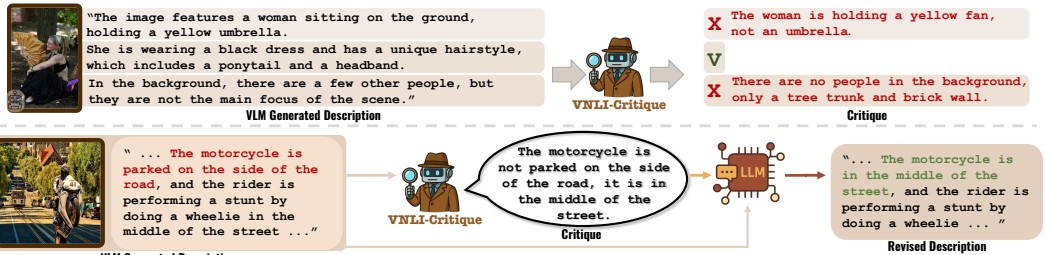

Figure 1: VNLI-Critique in action: operating as a Critic and within the Critic-and-Revise pipeline. (Top) As a Critic, VNLI-Critique evaluates sentence-level factuality in VLM captions and generates error explanations. (Bottom) In the pipeline, its critique of an incorrect sentence guides an LLM to revise it, demonstrating automated evaluation and correction of detailed captions.

## ABSTRACT

Large Vision-Language Models (VLMs) now generate highly detailed, paragraph-length image captions, yet evaluating their factual accuracy remains challenging. Current methods often miss fine-grained errors, being designed for shorter texts or lacking datasets with verified inaccuracies. We introduce *DOCCI-Critique*, a benchmark with 1,400 VLM-generated paragraph captions (100 images, 14 VLMs) featuring over 10,216 sentence-level human annotations of factual correctness and explanatory rationales for errors, all within paragraph context. Building on this, we develop *VNLI-Critique*, a model for automated sentence-level factuality classification and critique generation. We highlight three key applications: (1) VNLI-Critique demonstrates robust generalization, validated by state-of-the-art performance on the M-HalDetect benchmark and strong results in CHOCO-LATE claim verification. (2) The VNLI-Critique driven AutoRater for DOCCI-Critique provides reliable VLM rankings, showing excellent alignment with human factuality judgments (e.g., 0.98 Spearman). (3) An innovative Critic-and-Revise pipeline, where critiques from VNLI-Critique guide LLM-based corrections, achieves substantial improvements in caption factuality (e.g., a 46% gain on DetailCaps-4870). Our work offers a crucial benchmark alongside practical tools, designed to significantly elevate the standards for fine-grained evaluation and foster the improvement of VLM image understanding.

Figure 2: Sentence-level factuality assessment by VNLI-Critique. Figure shows an image, VLM-generated caption, and factuality judgments ($V$ Correct / $X$ Incorrect), illustrating VNLI-Critique's fine-grained, human-aligned fact-checking compared to zero-shot VLMs. Errors highlighted in red.

# 1 INTRODUCTION

Automatic descriptive image captioning, a prominent vision-language research area (Stefanini et al., 2023), has evolved from short highlights (Saouabe et al., 2023) to detailed, paragraph-length descriptions, thanks to powerful Large Vision-Language Models (LVLMs) (Chen et al., 2025; Dai et al., 2023; Li et al., 2024; Steiner et al., 2024; Wang et al., 2024b; Ye et al., 2024a). Evaluating these complex captions remains challenging; current metrics, often for short texts, miss subtle, fine-grained details and typically assess sentences in isolation, lacking crucial paragraph context for resolving ambiguities and co-references. While some studies address full paragraphs as Dong et al. (2024), granular sentence-level assessment is still difficult.

Existing VLM factuality benchmarks (e.g., for error/hallucination) often target short sentences or QA, inadequately addressing paragraph-length descriptions. While valuable, datasets like M-HalDetect (Gunjal et al., 2024) and CHOCOLATE (Huang et al., 2024) provide sentence-level annotations but may lack the error diversity of long-form VLM outputs or the consistent, detailed textual explanations for inaccuracies vital for fine-grained automated evaluation. Response-level evaluations like CAPTURE (Dong et al., 2024) score paragraphs, missing sentence-level granularity. A benchmark is critically needed, providing comprehensive, context-aware sentence-level factuality annotations, including explanatory rationales for errors, across diverse VLM-generated paragraphs.

To address this, we introduce *DOCCI-Critique*, a novel benchmark for fine-grained evaluation of detailed image descriptions. It comprises 1,400 paragraph-length captions (14 VLMs, 100 images), with its core value in 10,216 sentence-level human annotations. Each sentence's factuality was judged by five annotators, with detailed textual rationales for every identified error. This multi-perspective annotation offers a new resource for in-depth VLM analysis, providing multiply-verified sentence-level judgments and error explanations for long captions, distinct from existing datasets.

Building on DOCCI-Critique, we developed VNLI-Critique, a model for automated sentence-level factuality classification (using paragraph context) and explanatory critique generation. This dual capability enables a novel Critic-and-Revise pipeline (Fig. 1): VNLI-Critique evaluates and generates a critique for an incorrect VLM-generated sentence, guiding an LLM to revise it. The utility of VNLI-Critique and this pipeline is shown via key results: (1) VNLI-Critique achieves state-of-the-art performance on external benchmarks Gunjal et al. (2024) (0.76 Macro-F1) and competitive results on Huang et al. (2024), demonstrating strong generalization. (2) On our benchmark, the VNLI-Critique powered DOCCI-Critique AutoRater shows VLM rankings with strong correlation to human judgments (Table 3). (3) The Critic-and-Revise pipeline significantly boosts factuality of incorrect sentences (e.g., by 46% on Dong et al. (2024), 51% on Singla et al. (2024)), confirmed by human evaluation. Collectively, these contributions deliver an essential benchmark and powerful methodologies, enabling more precise fine-grained assessment and demonstrably enhancing the factual accuracy of detailed image understanding by VLMs.

# 2 RELATED WORK

Our work intersects with advancements in Vision-Language Models (VLMs) for detailed captioning, the development of image captioning datasets, and methodologies for evaluating caption quality, especially factual accuracy and fine-grained detail.

**Vision-Language Models**   Recent VLMs (Steiner et al., 2024; OpenAI et al., 2024; Bai et al., 2025; Li et al., 2024; Liu et al., 2024; Wang et al., 2024b; Deitke et al., 2024; Ye et al., 2024b; Chen et al., 2023; Lin et al., 2024; Dai et al., 2023; Zhu et al., 2023; Team et al., 2024) have achieved remarkable SOTA performance in multimodal tasks . Typically, they combine a visual encoder Radford et al. (2021); Zhai et al. (2023); Tschannen et al. (2025); Dosovitskiy et al. (2021) with an LLM (Yang et al., 2024; Chung et al., 2024; Chiang et al., 2023; Touvron et al., 2023), often using a connector module to bridge visual features with textual tokens. Training usually involves pre-training the visual encoder and then fine-tuning the LLM, with objectives like masked language modeling or image-text matching. While end-to-end training is explored, this two-stage approach is common, balancing dataset scale, computational resources, and evaluation.

**Image Captioning Datasets**   Image understanding and captioning datasets are crucial for advancing image captioning techniques. Early datasets like COCO (Lin et al., 2014), Flickr8k (Hodosh et al., 2013), and Flickr30k (Young et al., 2014) offered short-sentence, positive image-text pairs, primarily focusing on main objects and scenes. More recent datasets offer image-text pairs with longer, more detailed descriptions. The DCI dataset (Urbanek et al., 2024), for instance, introduces long, mask-aligned descriptions to evaluate VLM understanding of distinct image regions. PixelProse (Singla et al., 2024) offers 16.9 million synthetically generated captions using Gemini-1.0-Pro-Vision (Team et al., 2024); however, their correctness is not guaranteed. The IIW (Garg et al., 2024) dataset uses a VLM to generate initial captions, then a human-in-the-loop framework to ensure high-quality positive image-text pairs. M-HalDetect (Gunjal et al., 2024) and CHOCOLATE (Huang et al., 2024) use various VLMs to caption images and charts, respectively, then use sentence-level human annotation to assess correctness. DetailCaps (Dong et al., 2024) uses GPT-4 (OpenAI et al., 2024) to assign a quality score (0-5) to existing synthetic captions. The DOCCI dataset (Onoe et al., 2024) is a valuable resource for training and evaluating VLMs, offering high-resolution images of diverse real-life scenes paired with detailed, fully human-written descriptions. We leverage DOCCI as the image foundation for our DOCCI-Critique benchmark, for which we gathered over 10,216 new sentence-level factuality annotations. Furthermore, DOCCI's commercial-friendly license and its use of original, author-taken images without people address significant attribution and privacy challenges common in web-scraped datasets.

**Evaluation of Detailed Captions**   Traditional metrics (e.g., BLEU (Papineni et al., 2002), METEOR (Banerjee & Lavie, 2005), CIDEr (Vedantam et al., 2015)) use n-gram overlap to compare captions to references, often missing semantic nuance crucial for paragraph-length text. Embedding-based methods (e.g., CLIPScore (Hessel et al., 2021), SIGLip (Zhai et al., 2023)) offer better semantic assessment but evaluate sentences in isolation, lacking the paragraph context needed to resolve ambiguities or co-references. QA-based metrics (e.g., VQAScore (Lin et al., 2025), TIFA (Hu et al., 2023), VQ$^2$ (Yarom et al., 2023), GECKO (Wiles et al., 2025)) assess understanding via QA but face scalability challenges for long narratives. Recent work like Gordon et al. (2024) provides detailed feedback for misalignments, but primarily for text-to-image and shorter textual inputs. Response-level evaluations like Dong et al. (2024) score paragraphs but lack sentence-level factuality details. Our work provides the needed fine-grained, context-aware sentence-level factuality evaluation, including rich, human-verified explanatory feedback for detailed, paragraph-length image captions.

## 3   THE DOCCI-CRITIQUE BENCHMARK: CONSTRUCTION, ANNOTATION, AND ANALYSIS

*DOCCI-Critique* is a novel benchmark for fine-grained factuality assessment of paragraph-level image descriptions. Its primary purpose is twofold: (1) providing a robust platform to assess SOTA captioning models' descriptive capabilities and factual accuracy, and (2) serving as a challenging testbed for automated image understanding and fact-checking systems (auto-raters).

Construction began with 100 diverse, high-resolution images from the DOCCI dataset's 'qual-dev' split (Onoe et al., 2024). For each, 14 SOTA Large Vision-Language Models (Table 2) generated detailed, paragraph-length descriptions, yielding 1,400 model-generated descriptions. This corpus deliberately captures a wide spectrum of stylistic variations, detail levels, and factual accuracies, from concise and factual to more verbose accounts that might introduce subtle inconsistencies (e.g., object misidentification).

Table 1: Illustrative example from the DOCCI-Critique benchmark, detailing sentence-level annotations for fine-grained factuality assessment, including rater judgments and explanatory rationales.

| Image | 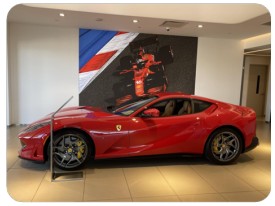 | | |
|---|---|---|---|
| **Description Sentence** | "... Behind the car, there is a large mural or poster on the wall ..." | "... The mural features a Formula 1 racing car, also red, with the number 16 prominently displayed on the side. ..." | "... The background of the mural includes a racing track with the colors of the French flag (blue, white, and red) and a checkered flag, indicating a racing theme ..." |
| **Does the sentence include a claim about the image? (Answers from 5 raters)** | ✅, ✅, ✅, ✅, ✅ | ✅, ✅, ✅, ✅, ✅ | ✅, ✅, ✅, ✅, ✅ |
| **Is the sentence factual? (Answers from 5 raters)** | ✅, ✅, ✅, ✅, ✅ | ✅, ✅, ❌, ❌, ✅ | ❌, ✅, ❌, ❌, ✅ |
| **Rationales** | - | • The 16 number is not on the side of the racing car, but in the front of it.
• The mural does feature a red Formula 1 race car, but the number 16 is painted on the front not the side | • There is no checkered flag visible in the image.
• There's no checkered flag in the poster/mural.
• The background mural does feature the colors blue, white and red, but there is no checkered flag |

The core of DOCCI-Critique is its rich human annotation layer. Following Steiner et al. (2024) (see Appendix for annotation template), five human annotators independently evaluated each sentence within the 1,400 descriptions for factual correctness against the image, assigning labels: 'Entailment' (factually supported), 'Neutral' (not verifiable/contradicted), 'Contradiction' (factually contradicted), or 'Nothing to assess' (e.g., filler). A sentence is classified as 'Non-entailment' if a majority labeled it 'Neutral' or 'Contradiction'. Crucially, annotators provided detailed textual rationales for each non-entailed judgment. Thus, a single non-entailed sentence often has multiple rationales, capturing diverse perspectives on the inaccuracy and offering insights into VLM error types (e.g., object misidentification, attribute/spatial errors, hallucination).

Table 1 illustrates this structure, showing per-sentence annotations: five independent factuality judgments (✅ for Entailment, ❌ for Contradiction/Neutral) and image content reliance. For non-entailed votes, collected textual rationales explain the specific error, as exemplified by the car mural's number placement or the non-existent checkered flag. Multiple rationales for one incorrect sentence reflect diverse annotator viewpoints.

This comprehensive annotation resulted in 10,216 sentence-level judgments. The dataset—with its diverse VLM outputs, fine-grained majority-vote factuality labels, and multiple rich explanatory rationales per error—is an invaluable resource. It enables rigorous VLM evaluation beyond surface-level similarity, allowing deeper probes into model understanding and descriptive fidelity. For more details on the human annotation process, including the task format and inter-rater agreement, please refer to Appendix C.

Table 2 details per-model statistics from DOCCI-Critique, including description and sentence lengths, factual accuracy, and lexical diversity. These internal statistics reveal quantitative and qualitative VLM behavioral differences. For instance, high-accuracy models like GPT-4o contrast with Gemini models that produce more sentences with comparable accuracy, suggesting a verbosity/detail vs. error-risk trade-off. Paragraphs in DOCCI-Critique average 752.7 characters in length. This is considerably longer than in other contemporary datasets used for factual error analysis, such as M-HalDetect (averaging 456.2 characters), CHOCOLATE (577.6), and DetailCaps-4870 (612.9). This emphasis on longer, more intricate descriptions, combined with patterns like common error types (discernible from rationales, though not directly in Table 1), highlights DOCCI-Critique's utility for nuanced comparative studies of VLM generation strategies and visual faithfulness.

Table 2: DOCCI-Critique statistics, detailing paragraph-level metrics and lexical diversity (unique 2-grams) for each LLM's generated image descriptions.

| | Description Length avg. | # Sentences Avg | Sentence Length Avg | % Correct Sentence in Description | Uni. 2-gram |
|---|---|---|---|---|---|
| MiniGPT-4 (Zhu et al., 2023) | 483.5 | 5.6 | 84.8 | 45.6 | 4,695 |
| mPLUG-Owl2-7B (Ye et al., 2024b) | 458.8 | 4.4 | 102.1 | 52.7 | 4,038 |
| LLaVa-1.5-7B (Liu et al., 2024) | 395.4 | 4.2 | 91.5 | 60.0 | 3,081 |
| InstructBLIP (Dai et al., 2023) | 509.8 | 4.0 | 195.4 | 61.3 | 3,260 |
| PALI-5B (Chen et al., 2023) | 1098.9 | 10.9 | 69.5 | 68.0 | 1,881 |
| VILA (Lin et al., 2024) | 870.7 | 8.6 | 100.4 | 78.1 | 6,841 |
| mPLUG-Owl3-7B (Ye et al., 2024a) | 118.0 | 2.0 | 65.2 | 80.4 | 700 |
| LLaVA-Onevision-7B (Li et al., 2024) | 672.0 | 6.4 | 107.7 | 81.8 | 5,878 |
| Molmo-7B-D (Deitke et al., 2024) | 747.6 | 6.6 | 111.9 | 82.7 | 6,788 |
| LLaVA-Onevision-7B-Chat (Li et al., 2024) | 1091.6 | 9.5 | 113.1 | 85.7 | 8,550 |
| Qwen2-VL-7B-Instruct (Wang et al., 2024a) | 1022.9 | 9.8 | 102.9 | 87.6 | 8,250 |
| Gemini-1.5-Pro (Team et al., 2024) | 1326.9 | 12.0 | 109.3 | 95.1 | 11,705 |
| Gemini-1.5-Flash (Team et al., 2024) | 1199.0 | 11.8 | 100.0 | 96.1 | 10,186 |
| GPT-4o [2024-08-06] (OpenAI et al., 2024) | 583.5 | 6.2 | 94.2 | 97.1 | 6,160 |
| TOTAL | 752.7 | 7.3 | 103.8 | 76.5 | 40,444 |

## 4 VNLI-CRITIQUE: DEVELOPMENT AND EVALUATION

### 4.1 VNLI-CRITIQUE MODEL DEVELOPMENT

We developed *VNLI-Critique* by fine-tuning the 10B parameter PaliGemma-2 architecture (Steiner et al., 2024) (details in Appendix) for automated sentence-level factuality assessment and critique generation. This required a specialized training dataset of VLM-generated captions, distinct from DOCCI-Critique, annotated for factuality and error critiques. To create this diverse training data, we first generated paragraph-length captions using over 70 PaliGemma-2 variants (fine-tuned with varied configurations on DOCCI training data (Onoe et al., 2024)) to capture many generation styles and potential errors. These synthetic captions were then human-annotated per the protocol in Section 3 (majority vote for labels; longest rationale for non-entailed sentences as critique target).

VNLI-Critique was fine-tuned on this curated data for two tasks using specific prompts incorporating paragraph context (<PREFIX>Claim-Prefix</PREFIX>), vital for accurate assessment of potentially ambiguous standalone sentences. For **Factuality Classification**, the prompt was: *"Given the image and the prompt prefix <PREFIX>Claim-Prefix</PREFIX>, does the following text align with the image: <TARGET>Target-Claim</TARGET>?"*, requiring a "Yes"/"No" prediction. For **Critique Generation**, the prompt was: *"Given the image and the prompt prefix <PREFIX>Claim-Prefix</PREFIX>, the text <TARGET>Target-Claim</TARGET> is considered inaccurate. Explain the misalignments and factual inaccuracies that make it inaccurate."*. This dual-task strategy enables VNLI-Critique to identify discrepancies and articulate their reasons.

### 4.2 FACTUALITY CLASSIFICATION: BENCHMARKING AND GENERALIZATION RESULTS

This section details the performance of VNLI-Critique in its factuality classification task, presenting key results from its application as an automated benchmarking tool on DOCCI-Critique and its generalization capabilities when tested on diverse external datasets.

**The DOCCI-Critique AutoRater: Automated VLM Benchmarking Results.** A primary application of VNLI-Critique's classification capability is to serve as an AutoRater for establishing an automated leaderboard that ranks Vision-Language Models (VLMs) based on their factual accuracy when describing images from the DOCCI-Critique benchmark. The objective is to provide a scalable and reliable alternative to extensive human evaluation for this task. To assess its viability, we evaluated VNLI-Critique alongside other VLM-based methods as potential automated rankers. We compared how well their automated assessments correlated with human judgments across three distinct factuality criteria: (1) Response-Level Correctness (whether the entire generated paragraph was factually accurate), (2) Percentage of Correct Sentences Overall (total correct sentences across all generated descriptions for a model), and (3) Average Percentage of Correct Sentences per Description. The detailed leaderboards showing the rankings of VLMs for each of these criteria, as

determined by both human evaluation and the automated methods, are provided in Appendix. The correlation results, using Spearman's $\rho$ (Sp $\rho$) and Kendall's $\tau$ (Kd $\tau$), are presented in Table 3. **VNLI-Critique demonstrates exceptional performance as an AutoRater, achieving the highest Spearman correlation with human rankings** on Response-Level Correctness (Sp $\rho$ = 0.981) and Percentage of Correct Sentences Overall (Sp $\rho$ = 0.979), and a very high correlation for Average Percentage of Correct Sentences per Description (Sp $\rho$ = 0.968). Its strong performance across these different evaluation granularities, significantly aligning with human assessments, validates its effectiveness as a reliable tool for automatically benchmarking VLM factuality on the DOCCI-Critique dataset.

**Evaluating Broader Applicability on External Benchmarks.** To assess VNLI-Critique's capabilities beyond our specific benchmark, we evaluate its performance on two established external datasets: M-HalDetect (Gunjal et al., 2024), a benchmark for detecting hallucinations in VLM descriptions of diverse images, and CHOCOLATE (Huang et al., 2024), which focuses on descriptions of charts and plots. We compare VNLI-Critique against various baselines, including other VLM-based classifiers and embedding-similarity methods, using two standard meta-evaluation metrics: ROC-AUC and Macro-F1. Figure 2 provides a qualitative example of these sentence-level classification comparisons.

The performance metrics reported in Table 4 were generated based on each model's output type. For models trained to classify via specific output tokens (e.g., 'Yes' and 'No') – this includes our VNLI-Critique and other VLM-based classifiers (where scores are derived via a 5-sample strategy) – metrics reflect both confidence and prediction. For all such VLM classifiers, the entailment score for ROC-AUC calculation is obtained by applying a softmax function to the confidence scores associated with the positive and negative classification outputs (yielding a normalized positive probability). The binary classification ('Accurate' or 'Inaccurate') for Macro-F1 is determined by selecting the label with the higher confidence score. In contrast, for methods like CLIPScore (Hessel et al., 2021), SigLIP (Zhai et al., 2023), and TIFA (Hu et al., 2023), which output numerical similarity scores, only ROC-AUC is reported, as it directly applies to such scores without requiring an arbitrary threshold.

As shown in Table 4, **VNLI-Critique achieves state-of-the-art (SOTA) performance on M-HalDetect. Furthermore, its highly competitive performance on the CHOCOLATE dataset demonstrates notable adaptability and robust reasoning capabilities**, even when evaluating descriptions of charts and graphs without specific training on such visual data. These strong findings across different benchmarks underscore the general utility of our fine-tuned model for factual verification tasks.

### 4.3 EVALUATING CRITIQUE GENERATION

Beyond classifying sentences' correctness, a key capability of VNLI-Critique is generating textual critiques that explain why a sentence is factually inaccurate. To assess the quality and correctness of these generated explanations, we conducted a dedicated human evaluation study. The evaluation proceeded as follows: First, we sampled sentences previously identified by human annotators as incorrect from our DOCCI-Critique and M-HalDetect benchmarks. For each, we prompted VNLI-

Table 3: Evaluating Automated Methods as AutoRaters. Correlation (Spearman's $\rho$, Kendall's $\tau$) between model-based rankings and human judgments of VLM factuality on DOCCI-Critique across three accuracy metrics. **Bold** indicates the best score, and underline indicates the second best.

| Ranking Method (Model) | % Response Correct | | % Sentences Overall | | % Sentences per Description | |
|---|---|---|---|---|---|---|
| | Sp $\rho$ | Kd $\tau$ | Sp $\rho$ | Kd $\tau$ | Sp $\rho$ | Kd $\tau$ |
| Emu3-Chat | -0.192 | -0.167 | 0.059 | 0.011 | 0.007 | -0.055 |
| InstructBLIP [Vicuna-7B] | -0.059 | -0.046 | 0.367 | 0.187 | 0.354 | 0.143 |
| Qwen2.5-VL-7B-Instruct | 0.290 | 0.249 | 0.692 | 0.516 | 0.697 | 0.495 |
| Janus-Pro-7B | 0.294 | 0.211 | 0.521 | 0.341 | 0.578 | 0.407 |
| mPLUG-Owl3-7B | 0.734 | 0.573 | 0.741 | 0.582 | 0.798 | 0.648 |
| LLaVa-OneVision[Qwen2-7B] | 0.889 | 0.760 | 0.855 | 0.758 | 0.851 | 0.736 |
| GPT-4o | 0.920 | 0.818 | 0.975 | 0.911 | **0.987** | **0.934** |
| Gemini-2.0-Flash | 0.972 | 0.884 | 0.976 | 0.911 | 0.956 | 0.890 |
| VNLI-Critique (Ours) | **0.981** | **0.928** | **0.979** | **0.912** | 0.968 | 0.906 |

Table 4: Evaluating VNLI-Critique's factuality classification: Comparison with baselines on in-distribution (DOCCI-CRITIQUE) and external (M-HalDetect, CHOCOLATE) datasets. Key results include SOTA on M-HalDetect and strong generalization to CHOCOLATE.

| Model | DOCCI-Critique | | M-HalDetect | | CHOCOLATE | |
|---|---|---|---|---|---|---|
| | ROC-AUC | Macro-F1 | ROC-AUC | Macro-F1 | ROC-AUC | Macro-F1 |
| CLIPScore | 0.48 | - | 0.59 | - | 0.56 | - |
| VQAScore [CLIP-FlanT5] | 0.73 | - | 0.79 | - | 0.71 | - |
| VQAScore [GPT-4o] | 0.88 | - | 0.85 | - | **0.84** | - |
| SigLIP | 0.50 | - | 0.63 | - | 0.56 | - |
| TIFA | 0.61 | - | 0.70 | - | 0.57 | - |
| PaliGemma2 [9B-448res] | 0.51 | 0.23 | 0.61 | 0.39 | 0.53 | 0.00 |
| Qwen2.5-VL-7B-Instruct | 0.65 | 0.36 | 0.81 | 0.75 | 0.81 | 0.74 |
| InstructBLIP [Vicuna-7B] | 0.50 | 0.45 | 0.45 | 0.40 | 0.53 | 0.37 |
| Emu3-Chat | 0.51 | 0.50 | 0.52 | 0.42 | 0.50 | 0.37 |
| Janus-Pro-7B | 0.67 | 0.58 | 0.72 | 0.59 | 0.65 | 0.47 |
| LLaVa-OneVision[Qwen2-7B] | 0.76 | 0.58 | 0.82 | 0.60 | 0.75 | 0.44 |
| mPLUG-Owl3-7B | 0.73 | 0.65 | 0.76 | 0.68 | 0.68 | 0.54 |
| Gemini-2.0-Flash | 0.73 | 0.74 | 0.74 | 0.74 | 0.81 | **0.79** |
| GPT-4o | - | 0.74 | - | 0.69 | - | 0.70 |
| VNLI-Critique (Ours) | **0.93** | **0.83** | **0.86** | **0.76** | 0.73 | 0.68 |

Critique and other competitive VLMs (Table 5) to generate an explanation detailing the specific misalignments, using the prompt format from Section 4.1. Human annotators were then presented with evaluation instances containing: (1) the original image, (2) the incorrect sentence, and (3) the model-generated critique. Annotators judged if the critique accurately and relevantly identified the factual error(s) based on the image. Table 5 reports the percentage of critiques deemed correct and relevant. The results (Table 5) demonstrate the effectiveness of our specialized, open-source VNLI-Critique (a 10B parameter model). Despite its significantly smaller scale, our model achieves critique generation quality comparable to, and in some cases exceeding, large-scale state-of-the-art commercial models. On DOCCI-Critique sentences, VNLI-Critique achieves the highest score (73.39%), slightly surpassing GPT-4o (73.1%). On M-HalDetect, it scores 79.33%, performing comparably to Gemini-2.0-Flash (79.89%) and again surpassing GPT-4o (78.77%). Notably, VNLI-Critique also significantly outperforms other open-source VLMs of a similar size, such as Janus-Pro-7B, Qwen-2.5-VL-Instruct, and LLaVA-OV on both datasets. This highlights VNLI-Critique's ability to generate high-quality, accurate explanations, providing a crucial capability for interpretable feedback and downstream correction that is accessible to the wider research community.

Table 5: Human evaluation of critique quality. Percentage of generated explanations judged as correct and relevant for incorrect sentences sampled from DOCCI-Critique and M-HalDetect.

| | DOCCI-Critique | M-HalDetect |
|---|---|---|
| LLaVA-OV | 35.96 | 48.04 |
| Qwen-2.5-VL-Instruct | 45.03 | 58.1 |
| Janus-Pro-7B | 44.15 | 62.57 |
| Gemini-2.0-Flash | 64.91 | **79.89** |
| GPT-4o | 73.1 | 78.77 |
| VNLI-Critique (Ours) | **73.39** | 79.33 |

## 5 CRITIC-AND-REVISE

Many vision-language tasks, from image captioning to text-to-image generation, rely heavily on large-scale datasets of image-text pairs, often utilizing VLM-generated captions for training or as part of their data. For example, large synthetic caption datasets like PixelProse (Singla et al., 2024) are used to train captioning models, and datasets pairing images with descriptive text are fundamental for training text-to-image synthesis models (e.g., leveraging datasets like Schuhmann et al. (2022)). However, the factual accuracy and visual alignment of these automatically generated or

web-crawled captions can vary, potentially introducing noise or inaccuracies into downstream model training. Improving the quality and factual alignment of such datasets is therefore crucial for advancing these fields. Leveraging the critique generation capability of VNLI-Critique, we introduce and evaluate a novel *Critic-and-Revise* pipeline. This pipeline is designed not only to correct individual factual inaccuracies in image captions but also offers a pathway to enhance the overall quality of image-text training datasets, thereby potentially improving the performance of models trained on them. This section first outlines the pipeline's methodology (Section 5.1). We then evaluate its applicability in correcting synthetically generated captions from large-scale datasets (Section 5.2).

## 5.1 PIPELINE METHODOLOGY

The Critic-and-Revise pipeline, illustrated in Figure 1, operates in two main steps. First, in the *Critic step*, VNLI-Critique analyzes each sentence of a given caption using its classification function; sentences identified as factually inaccurate trigger the generation of a textual critique explaining the specific error based on the image content. Subsequently, in the *Revise step*, the original inaccurate sentence and its corresponding critique from VNLI-Critique are used to instruct a separate Large Language Model (LLM) to fix the inaccurate description. For our experiments, we utilized Gemini-2.0-Flash[1] as the revision LLM. This revision LLM is prompted to rewrite the original sentence, specifically addressing the factual errors highlighted in the critique, while aiming to preserve relevant information and maintain stylistic coherence. The full Critic-and-Revise cycle—factuality classification by VNLI-Critique for all sentences, followed by critique-guided revision for those flagged as inaccurate—produces a revised caption with enhanced factual alignment to the image.

## 5.2 CORRECTING SYNTHETICALLY GENERATED CAPTIONS

To demonstrate the downstream utility of our proposed Critic-and-Revise pipeline, we applied it to captions from two large-scale datasets known for detailed yet potentially unverified descriptions: PixelProse (Singla et al., 2024), featuring 16.9M synthetic caption pairs, and DetailCaps-4870 (Dong et al., 2024), a subset with 4,870 images each accompanied by three detailed synthetic captions. We conducted a human study to evaluate the pipeline's effectiveness: after VNLI-Critique identified and critiqued inaccurate sentences, and the revision LLM corrected them, human evaluators assessed the factual correctness of both the original flagged sentences (for critic precision) and the revised sentences (for pipeline effectiveness).

The results, summarized in Table 6, highlight significant improvements. For DetailCaps-4870, while VNLI-Critique's initial flagging showed a 15% false positive rate (original sentences deemed correct by humans), **the pipeline successfully corrected a large portion of the genuinely inaccurate sentences,** with human judges confirming 61% of the revised sentences as factually accurate. This represents a 46% gain in accuracy for the set of sentences initially considered incorrect by the critic. VNLI-Critique's own re-evaluation classified 64% of these revised sentences as accurate, indicating strong self-consistency. Similar positive trends were observed for PixelProse, where human judges found 75% of revised sentences to be accurate (a 51% gain), demonstrating the pipeline's capability to enhance factual accuracy in detailed image captions at scale. Qualitative examples illustrating the Critic-and-Revise process, including original incorrect sentences, critiques from VNLI-Critique, and the LLM-revised sentences, are provided in Appendix.

Table 6: Critic-and-Revise Pipeline factuality: Human and VNLI-Critique judgments on original vs. revised claims. $\Delta$ = accuracy increase post-revision. **The pipeline markedly improves claim accuracy (human-confirmed to 61% on DetailCaps, 75% on PixelProse for fixed claims, from low initial values).** VNLI-Critique's judgments align, showing high self-consistency.

| | DetailCaps-4870 | | | PixelProse | | |
|---|---|---|---|---|---|---|
| **Judge Type** | **Original** | **Fixed** | $\Delta$ | **Original** | **Fixed** | $\Delta$ |
| Human Judge | 15% | 61% | +46% | 24% | 75% | +51% |
| VNLI-Critique as Judge | 0% | 64% | +64% | 0 | 61% | +61% |

---

[1]Accessed via the Vertex AI API: `https://cloud.google.com/vertex-ai`

## 6    LIMITATIONS AND FUTURE WORK

The DOCCI-Critique benchmark, while richly annotated with 1,400 VLM-generated captions and over 10,000 sentence judgments, is constructed from a base set of 100 unique images. While these images provide diversity and the caption variations are extensive, expanding the number of base images could further enhance the benchmark's statistical power and coverage. Nevertheless, our experiments demonstrate strong generalization capabilities. Specifically, VNLI-Critique, when trained for factuality classification on DOCCI-Critique, performs well on external, unseen claim verification datasets like M-HalDetect and CHOCOLATE (Section 4.2). Furthermore, our Critic-and-Revise pipeline, leveraging critiques from VNLI-Critique, effectively corrects captions on entirely different datasets such as DetailCaps-4870 and PixelProse (Section 5.2). This collective evidence of generalization across different tasks and datasets suggests the current DOCCI-Critique size is effective for developing robust and transferable evaluation models and correction methodologies.

Additionally, while VNLI-Critique achieves strong results in several settings, its performance is not perfect. Our pipeline evaluation (Section 5) indicates that its factuality classification can result in false positives and false negatives (e.g., a 15% false positive rate on DetailCaps-4870). The quality of its generated critiques, while generally high as shown by human evaluations in Table 5, can also exhibit variability. Future work could enhanceVNLI-Critique's performance by further leveraging our rich annotations. For example, one could investigate methods to merge multiple rationales provided by different annotators into a single, more comprehensive and concise explanation, instead of solely using the longest rationale as the training target for critique generation. Furthermore, our annotation protocol captures whether a sentence's factuality is dependent on the image content or relies on world knowledge (e.g., distinguishing "The cat is on the mat" which requires the image, from "A cat is a mammal" which does not). This currently unused label could enable a two-stage verification process: first classifying if image grounding is needed, and then applying either the visual fact-checker (VNLI-Critique) or a knowledge-based verifier accordingly, potentially improving overall accuracy and efficiency.

Regarding the Critic-and-Revise pipeline, its current design involves two distinct steps: critique generation by VNLI-Critique followed by revision using a separate LLM. While effective, this contrasts with a hypothetical end-to-end model that might directly output a corrected sentence. However, we argue that the two-step approach offers significant advantages in interpretability. Generating an explicit critique allows for a clear understanding of why a sentence was flagged and what specific error is being addressed. This insight into error types and sources is valuable for analyzing and improving the underlying captioning models, a benefit potentially lost in a direct, black-box correction approach. Therefore, while future work might explore direct revision models, the explanatory power of the intermediate critique remains a key strength of our pipeline.

Addressing these limitations and exploring the suggested avenues for leveraging the full extent of the dataset annotations offers exciting directions for future research in robust and interpretable evaluation of detailed image understanding.

## 7    CONCLUSION

This work tackled the critical challenge of evaluating and improving the factual accuracy of detailed, paragraph-length VLM-generated image captions. We introduced *DOCCI-Critique*, a novel benchmark featuring 1,400 VLM captions with over 10,216 sentence-level human annotations for factuality, including explanatory rationales for errors, providing a vital resource for fine-grained VLM assessment. Building on this, we developed VNLI-Critique, a model proficient in automated factuality classification and critique generation. VNLI-Critique demonstrated strong generalization with state-of-the-art results on external datasets like M-HalDetect, and its use in the DOCCI-Critique showed high correlation with human judgments (0.98 Spearman). Furthermore, we presented a novel Critic-and-Revise pipeline where VNLI-Critique's critiques guide an LLM to automatically correct factual errors, significantly improving caption accuracy as confirmed by human evaluation. Collectively, DOCCI-Critique, VNLI-Critique, and the Critic-and-Revise pipeline offer essential tools and methodologies for advancing VLMs towards generating more detailed, fluent, and factually reliable image descriptions. Future work, as outlined in Section 6, will explore expanding the benchmark and further enhancing the pipeline's capabilities.

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
