# A DOCCI-CRITIQUE AUTORATER LEADERBOARDS

This appendix presents the complete leaderboards detailing the performance of various Vision-Language Models (VLMs) as automated rankers (AutoRaters) on the DOCCI-Critique benchmark. These tables supplement the summary correlation metrics (Spearman's $\rho$ and Kendall's $\tau$) found in Section 4.2, Table 3. Our goal was to assess how well automated methods, including our VNLI-Critique, rank caption-generating VLMs by factual accuracy against human-derived Ground Truth rankings.

Three leaderboards are provided, each for a distinct factuality criterion:

1. **Response-Level Correctness:** Percentage of entirely factually accurate paragraphs (Table 7).

2. **Correct Sentences Overall:** Total percentage of correct sentences across all descriptions (Table 9).

3. **Correct Sentences per Description:** Average percentage of correct sentences per description (Table 8).

In each leaderboard (Tables 7, 9, and 8), rows list the 14 caption-generating VLMs from DOCCI-Critique (details in Table 2). Columns denote automated ranking methods (e.g., 'Ours (VNLI-Critique)', 'GPT-4o'). Cells show the rank assigned by the column's method to the row's VLM, with the superscript indicating the raw metric score. The final two rows report Spearman's $\rho$ (with p-value superscript) and Kendall's $\tau$ correlations against the Ground Truth for that criterion, offering a nuanced view of each AutoRater's performance.

Table 7: VLM AutoRater rankings for Response-Level Correctness on DOCCI-Critique. Each cell shows $Rank^{Metric-Score}$. Final two rows: Spearman's $\rho^{p-value}$ and Kendall's $\tau^{p-value}$ correlation against Human.

| Captioner VLM | Human | Ours | Gemini 2.0-Flash | GPT-4o | InstructBLIP | LLaVa-OV | Janus-Pro-7B | Qwen2.5-VL | mPLUG-Owl3-7B | Emu3-Chat |
|---|---|---|---|---|---|---|---|---|---|---|
| MiniGPT-4 | $14^{0.04}$ | $14^{0.06}$ | $14^{0.09}$ | $14^{0.12}$ | $7^{0.96}$ | $14^{0.42}$ | $13^{0.35}$ | $4^{0.00}$ | $13^{0.11}$ | $3^{0.93}$ |
| MPlugOwl-2 | $13^{0.11}$ | $12^{0.12}$ | $13^{0.28}$ | $12^{0.18}$ | $2^{0.98}$ | $12^{0.56}$ | $10^{0.56}$ | $4^{0.00}$ | $12^{0.17}$ | $9^{0.55}$ |
| LLaVA | $11^{0.19}$ | $11^{0.17}$ | $9^{0.40}$ | $11^{0.26}$ | $3^{0.97}$ | $10^{0.71}$ | $5^{0.68}$ | $4^{0.00}$ | $8^{0.21}$ | $12^{0.48}$ |
| PALI-5B | $12^{0.11}$ | $13^{0.09}$ | $12^{0.31}$ | $13^{0.17}$ | $9^{0.96}$ | $11^{0.58}$ | $14^{0.19}$ | $4^{0.00}$ | $14^{0.07}$ | $2^{0.99}$ |
| VILA | $10^{0.21}$ | $10^{0.20}$ | $10^{0.39}$ | $9^{0.33}$ | $13^{0.91}$ | $9^{0.85}$ | $1^{0.79}$ | $4^{0.00}$ | $8^{0.21}$ | $4^{0.78}$ |
| InstructBLIP | $9^{0.26}$ | $9^{0.21}$ | $11^{0.38}$ | $10^{0.31}$ | $9^{0.96}$ | $13^{0.54}$ | $9^{0.60}$ | $2^{0.01}$ | $6^{0.26}$ | $10^{0.54}$ |
| Molmo-7B-D | $8^{0.31}$ | $8^{0.27}$ | $7^{0.68}$ | $8^{0.58}$ | $1^{0.99}$ | $5^{0.93}$ | $4^{0.70}$ | $4^{0.00}$ | $8^{0.21}$ | $4^{0.78}$ |
| LLaVA-OV-7B-Chat | $7^{0.36}$ | $6^{0.35}$ | $8^{0.63}$ | $5^{0.64}$ | $12^{0.93}$ | $8^{0.90}$ | $8^{0.66}$ | $4^{0.00}$ | $4^{0.30}$ | $8^{0.58}$ |
| Qwen2-VL-7B-Instruct | $5^{0.41}$ | $4^{0.45}$ | $5^{0.75}$ | $4^{0.65}$ | $3^{0.97}$ | $6^{0.91}$ | $2^{0.78}$ | $4^{0.00}$ | $5^{0.29}$ | $11^{0.49}$ |
| LLaVA-OV-7B | $5^{0.41}$ | $7^{0.34}$ | $6^{0.72}$ | $7^{0.63}$ | $14^{0.90}$ | $2^{0.97}$ | $6^{0.67}$ | $2^{0.01}$ | $2^{0.41}$ | $7^{0.61}$ |
| Gemini-1.5-Pro | $4^{0.64}$ | $5^{0.43}$ | $4^{0.84}$ | $3^{0.67}$ | $3^{0.97}$ | $6^{0.91}$ | $12^{0.49}$ | $4^{0.00}$ | $7^{0.24}$ | $14^{0.31}$ |
| Gemini-1.5-Flash | $3^{0.68}$ | $3^{0.52}$ | $3^{0.88}$ | $5^{0.64}$ | $11^{0.94}$ | $4^{0.94}$ | $11^{0.51}$ | $4^{0.00}$ | $11^{0.18}$ | $13^{0.40}$ |
| mPLUG-Owl3-7B | $2^{0.71}$ | $2^{0.69}$ | $2^{0.93}$ | $2^{0.77}$ | $7^{0.96}$ | $1^{0.98}$ | $3^{0.73}$ | $1^{0.02}$ | $1^{0.75}$ | $1^{1.00}$ |
| GPT-4o[2024-08-06] | $1^{0.83}$ | $1^{0.73}$ | $1^{0.94}$ | $1^{0.89}$ | $2^{0.97}$ | $2^{0.97}$ | $6^{0.67}$ | $4^{0.00}$ | $3^{0.37}$ | $6^{0.74}$ |
| Spearman's Rank $\rho$ | – | $\mathbf{0.98}^{5e-10}$ | $0.97^{6e-9}$ | $0.92^{3e-6}$ | $-0.06^{8e-1}$ | $0.88^{2e-5}$ | $0.30^{3e-1}$ | $0.29^{3e-1}$ | $0.73^{3e-3}$ | $-0.2^{5e-1}$ |
| Kendall Tau $\tau$ | – | $\mathbf{0.93}^{4e-6}$ | $\underline{0.89}^{1e-5}$ | $0.82^{5e-5}$ | $-0.05^{8e-1}$ | $0.76^{2e-4}$ | $0.21^{3e-1}$ | $0.25^{3e-1}$ | $0.27^{5e-3}$ | $-0.17^{4e-1}$ |

Table 8: VLM AutoRater rankings for Average Percentage of Correct Sentences on DOCCI-Critique. Each cell shows $Rank^{Metric-Score}$. Final two rows: Spearman's $\rho^{p-value}$ and Kendall's $\tau^{p-value}$ correlation against Human.

| Captioner VLM | Human | Ours | Gemini 2.0-Flash | GPT-4o | InstructBLIP | LLaVa-OV | Janus-Pro-7B | Qwen2.5-VL | mPLUG-Owl3-7B | Emu3-Chat |
|---|---|---|---|---|---|---|---|---|---|---|
| MiniGPT-4 | $14^{0.46}$ | $14^{0.48}$ | $14^{0.53}$ | $14^{0.42}$ | $8^{0.99}$ | $14^{0.84}$ | $13^{0.77}$ | $13^{0.05}$ | $14^{0.51}$ | $3^{0.99}$ |
| MPlugOwl-2 | $13^{0.53}$ | $13^{0.54}$ | $13^{0.66}$ | $13^{0.56}$ | $7^{0.99}$ | $12^{0.89}$ | $12^{0.86}$ | $11^{0.05}$ | $13^{0.53}$ | $12^{0.86}$ |
| LLaVA | $12^{0.60}$ | $11^{0.59}$ | $11^{0.75}$ | $12^{0.59}$ | $9^{0.99}$ | $11^{0.91}$ | $9^{0.90}$ | $9^{0.08}$ | $11^{0.58}$ | $14^{0.83}$ |
| InstructBLIP | $11^{0.61}$ | $12^{0.58}$ | $12^{0.71}$ | $11^{0.62}$ | $12^{0.98}$ | $13^{0.86}$ | $10^{0.89}$ | $10^{0.05}$ | $12^{0.56}$ | $13^{0.84}$ |
| PALI-5B | $10^{0.67}$ | $10^{0.68}$ | $10^{0.79}$ | $10^{0.79}$ | $3^{1.00}$ | $10^{0.91}$ | $14^{0.73}$ | $12^{0.05}$ | $10^{0.61}$ | $2^{1.00}$ |
| VILA | $9^{0.78}$ | $8^{0.78}$ | $9^{0.86}$ | $9^{0.81}$ | $11^{0.99}$ | $9^{0.98}$ | $1^{0.97}$ | $4^{0.22}$ | $8^{0.76}$ | $4^{0.96}$ |
| mPLUG-Owl3-7B | $8^{0.80}$ | $6^{0.80}$ | $4^{0.98}$ | $8^{0.87}$ | $13^{0.98}$ | $5^{0.99}$ | $11^{0.87}$ | $14^{0.05}$ | $2^{0.85}$ | $1^{1.00}$ |
| LLaVA-OV-7B | $7^{0.82}$ | $6^{0.80}$ | $7^{0.94}$ | $6^{0.91}$ | $14^{0.97}$ | $1^{1.00}$ | $5^{0.94}$ | $6^{0.20}$ | $7^{0.81}$ | $8^{0.92}$ |
| Molmo-7B-D | $6^{0.83}$ | $9^{0.78}$ | $6^{0.95}$ | $7^{0.90}$ | $1^{1.00}$ | $7^{0.99}$ | $4^{0.94}$ | $8^{0.10}$ | $9^{0.73}$ | $5^{0.95}$ |
| LLaVA-OV-7B-Chat | $5^{0.86}$ | $5^{0.85}$ | $8^{0.94}$ | $4^{0.94}$ | $10^{0.99}$ | $8^{0.99}$ | $3^{0.94}$ | $1^{0.28}$ | $1^{0.85}$ | $7^{0.93}$ |
| Qwen2-VL-7B-Instruct | $4^{0.88}$ | $4^{0.88}$ | $5^{0.97}$ | $5^{0.93}$ | $4^{1.00}$ | $6^{0.99}$ | $2^{0.97}$ | $5^{0.22}$ | $5^{0.83}$ | $9^{0.91}$ |
| Gemini-1.5-Pro | $3^{0.95}$ | $3^{0.93}$ | $3^{0.99}$ | $2^{0.97}$ | $2^{1.00}$ | $4^{0.99}$ | $8^{0.93}$ | $3^{0.25}$ | $4^{0.83}$ | $11^{0.89}$ |
| Gemini-1.5-Flash | $2^{0.96}$ | $2^{0.94}$ | $2^{0.99}$ | $3^{0.95}$ | $5^{0.99}$ | $3^{0.99}$ | $7^{0.93}$ | $2^{0.27}$ | $3^{0.84}$ | $10^{0.90}$ |
| GPT-4o[2024-08-06] | $1^{0.97}$ | $1^{0.95}$ | $1^{0.99}$ | $1^{0.98}$ | $6^{0.99}$ | $2^{1.00}$ | $6^{0.93}$ | $7^{0.17}$ | $6^{0.82}$ | $6^{0.95}$ |
| Spearman's Rank $\rho$ | – | $\underline{0.97}^{1e-8}$ | $0.96^{9e-8}$ | $\mathbf{0.97}^{7e-11}$ | $0.35^{2e-1}$ | $0.85^{1e-4}$ | $0.58^{3e-2}$ | $0.70^{5e-3}$ | $0.80^{6e-4}$ | $0.00^{1e-0}$ |
| Kendall Tau $\tau$ | – | $\underline{0.91}^{7e-6}$ | $0.90^{2e-7}$ | $\mathbf{0.93}^{1e-8}$ | $0.14^{5e-1}$ | $0.74^{7e-5}$ | $0.40^{4e-2}$ | $0.50^{1e-2}$ | $0.65^{7e-4}$ | $-0.05^{8e-1}$ |

Table 9: VLM AutoRater rankings for Percentage of Correct Sentences Overall on DOCCI-Critique. Each cell shows $Rank^{Metric-Score}$. Final two rows: Spearman's $\rho^{p-value}$ and Kendall's $\tau^{p-value}$ correlation against Human.

| Captioner VLM | Human | Ours | Gemini 2.0-Flash | GPT-4o | InstructBLIP | LLaVa-OV | Janus-Pro-7B | Qwen2.5-VL | mPLUG-Owl3-7B | Emu3-Chat |
|---|---|---|---|---|---|---|---|---|---|---|
| MiniGPT-4 | $14^{0.48}$ | $14^{0.49}$ | $14^{0.55}$ | $14^{0.44}$ | $7^{0.99}$ | $14^{0.83}$ | $13^{0.77}$ | $11^{0.06}$ | $13^{0.52}$ | $3^{0.99}$ |
| MPlugOwl-2 | $13^{0.52}$ | $13^{0.53}$ | $13^{0.64}$ | $13^{0.56}$ | $7^{0.99}$ | $12^{0.87}$ | $11^{0.85}$ | $13^{0.05}$ | $14^{0.51}$ | $12^{0.86}$ |
| InstructBLIP | $12^{0.57}$ | $12^{0.57}$ | $12^{0.68}$ | $11^{0.59}$ | $11^{0.99}$ | $13^{0.84}$ | $10^{0.87}$ | $12^{0.05}$ | $12^{0.54}$ | $14^{0.82}$ |
| LLaVA | $11^{0.59}$ | $11^{0.59}$ | $10^{0.74}$ | $12^{0.59}$ | $7^{0.99}$ | $10^{0.90}$ | $9^{0.89}$ | $9^{0.08}$ | $11^{0.58}$ | $13^{0.83}$ |
| PALI-5B | $10^{0.67}$ | $10^{0.66}$ | $10^{0.74}$ | $10^{0.62}$ | $4^{1.00}$ | $11^{0.88}$ | $14^{0.73}$ | $10^{0.07}$ | $10^{0.59}$ | $2^{1.00}$ |
| VILA | $9^{0.79}$ | $8^{0.79}$ | $9^{0.86}$ | $9^{0.81}$ | $11^{0.99}$ | $9^{0.98}$ | $1^{0.97}$ | $4^{0.22}$ | $8^{0.77}$ | $4^{0.96}$ |
| LLaVA-OV-7B | $8^{0.82}$ | $7^{0.81}$ | $8^{0.95}$ | $6^{0.91}$ | $13^{0.98}$ | $1^{1.00}$ | $5^{0.93}$ | $6^{0.21}$ | $2^{0.85}$ | $9^{0.91}$ |
| mPLUG-Owl3-7B | $7^{0.82}$ | $6^{0.82}$ | $4^{0.96}$ | $8^{0.84}$ | $14^{0.98}$ | $8^{0.99}$ | $12^{0.81}$ | $14^{0.05}$ | $3^{0.84}$ | $1^{1.00}$ |
| Molmo-7B-D | $6^{0.83}$ | $9^{0.95}$ | $6^{0.95}$ | $7^{0.90}$ | $1^{1.00}$ | $6^{0.99}$ | $4^{0.94}$ | $8^{0.10}$ | $9^{0.72}$ | $5^{0.96}$ |
| Qwen2-VL-7B-Instruct | $5^{0.87}$ | $4^{0.88}$ | $4^{0.96}$ | $5^{0.93}$ | $2^{1.00}$ | $5^{0.99}$ | $2^{0.97}$ | $5^{0.22}$ | $6^{0.83}$ | $8^{0.91}$ |
| LLaVA-OV-7B-Chat | $4^{0.87}$ | $5^{0.88}$ | $6^{0.95}$ | $4^{0.94}$ | $10^{0.99}$ | $7^{0.99}$ | $3^{0.96}$ | $1^{0.31}$ | $1^{0.87}$ | $7^{0.92}$ |
| Gemini-1.5-Pro | $3^{0.95}$ | $3^{0.93}$ | $3^{0.99}$ | $2^{0.97}$ | $2^{1.00}$ | $4^{0.99}$ | $8^{0.93}$ | $3^{0.25}$ | $5^{0.84}$ | $11^{0.89}$ |
| Gemini-1.5-Flash | $2^{0.96}$ | $2^{0.94}$ | $1^{0.99}$ | $3^{0.95}$ | $5^{1.00}$ | $1^{1.00}$ | $5^{0.93}$ | $2^{0.27}$ | $4^{0.84}$ | $9^{0.91}$ |
| GPT-4o[2024-08-06] | $1^{0.97}$ | $1^{0.95}$ | $1^{0.99}$ | $1^{0.98}$ | $5^{1.00}$ | $1^{1.00}$ | $5^{0.93}$ | $7^{0.16}$ | $7^{0.81}$ | $6^{0.95}$ |
| Spearman's Rank $\rho$ | – | $\mathbf{0.98}^{1e-9}$ | $0.98^{2e-9}$ | $0.97^{2e-9}$ | $0.37^{2e-1}$ | $0.86^{10e-5}$ | $0.52^{6e-2}$ | $0.69^{6e-3}$ | $0.74^{2e-3}$ | $0.06^{8e-1}$ |
| Kendall Tau $\tau$ | – | $\mathbf{0.91}^{5e-8}$ | $\underline{0.91}^{5e-8}$ | $0.91^{5e-8}$ | $0.19^{4e-1}$ | $0.76^{4e-5}$ | $0.34^{1e-1}$ | $0.52^{10e-3}$ | $0.58^{3e-3}$ | $0.01^{1e+0}$ |

# B  VNLI-Critique Model Development Details

This section provides further details on the architecture, fine-tuning process, and computational resources utilized for the development of our VNLI-Critique model, as introduced in Section 4 of the main paper.

## B.1  Model Architecture

VNLI-Critique is developed by fine-tuning the PaliGemma 10B architecture Steiner et al. (2024). This architecture integrates a Gemma2-9B Large Language Model (LLM) Rivière et al. (2024) as its textual backbone and a SigLIP model Zhai et al. (2023) as its visual encoder. For visual processing, input images are standardized to a resolution of $448px^2$ pixels. At this resolution, the SigLIP visual encoder processes each image into a sequence of 1024 visual tokens, which are subsequently fed into the LLM component for multimodal understanding and generation tasks.

## B.2  Fine-tuning Procedure

We performed full fine-tuning of the PaliGemma 10B model to develop VNLI-Critique. The fine-tuning process was conducted for 5 epochs. A batch size of 128 was used, with a dropout rate of 0.1 applied to aid regularization. No weight decay was utilized during training. The Adam optimizer Kingma & Ba (2015) was employed with its default hyperparameters, and a constant learning rate of $1 \times 10^{-6}$ was maintained throughout the fine-tuning process.

## B.3  Computational Resources

The training of the VNLI-Critique model was executed on Google Cloud TPUv5e Google Cloud (20xx) accelerators. Specifically, a configuration of 128 TPUv5e chips was utilized for the fine-tuning task. The total training time for the 5 epochs was approximately 1 hour and 30 minutes. Based on an estimated cost of \$1.20 per chip-hour, the total computational cost for training VNLI-Critique was approximately \$230.40.

# C  Human Annotation Details

The creation of the DOCCI-Critique benchmark and the evaluation of our models' outputs, including critique generation and the Critic-and-Revise pipeline, relied on comprehensive human annotations. We engaged third-party human annotators sourced through Prolific[2]. Each data entry subject to human evaluation, whether for sentence-level factuality in DOCCI-Critique or for the quality assessment of generated critiques, was independently assessed by five different annotators. This

---

[2] https://www.prolific.com/

multi-annotator approach helps ensure robustness and mitigate individual biases in the collected judgments. Annotators were compensated at a rate of $20 per hour for their work.

This same rigorous 5-annotator protocol was used for annotating both the DOCCI-Critique benchmark (comprising 10,216 sentence-level judgments and the training set for VNLI-Critique (comprising 75,363 sentence-level annotations). To quantify annotation quality for the benchmark, we computed Fleiss' Kappa for factual correctness and achieved a score of 0.48. We interpret this as moderate agreement, reflecting the highly nuanced nature of fine-grained factual assessment.

The following subsections provide an illustrative overview of the annotation interfaces designed for the two primary human evaluation tasks: assessing the factuality of VLM-generated description sentences (Section C.1) and evaluating the quality of generated critiques (Section C.2). We also provide a detailed analysis of the error categories found in our dataset (Section C.3).

## C.1 DESCRIPTION SENTENCES ANNOTATION INTERFACE

For the task of annotating sentence-level factuality within VLM-generated paragraph descriptions (as detailed in Section 3 for the DOCCI-Critique benchmark), annotators were presented with an interface displaying the source image, the full paragraph context, and the specific sentence under evaluation. Figure 3 illustrates a representative example of this annotation interface. Annotators were asked to judge whether the sentence accurately described the image content, providing labels such as 'Entailment', 'Neutral', or 'Contradiction', and to supply textual rationales for any non-entailed judgments.

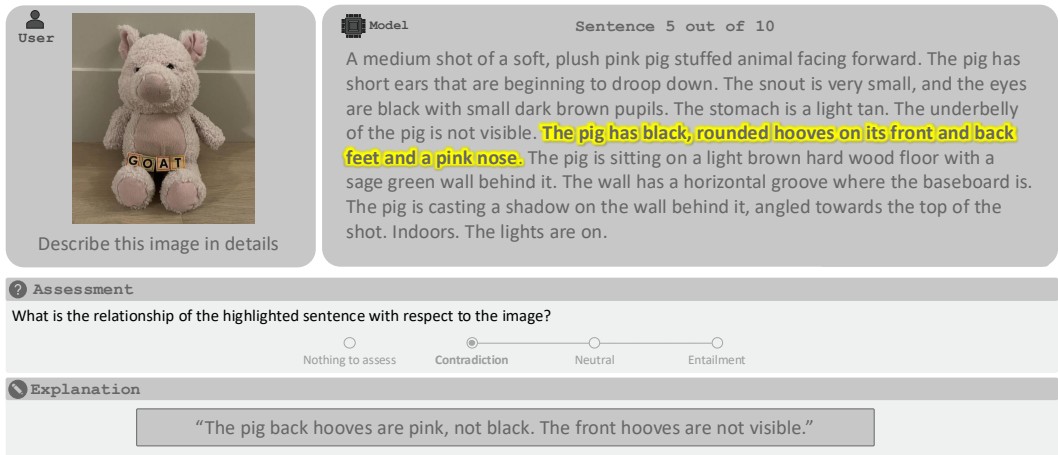

Figure 3: Example of the Description Sentences Annotation Interface. Annotators are shown the image, the full VLM-generated paragraph, and a highlighted sentence. They assess its factuality by selecting a label (here, 'Contradiction') and providing a textual explanation for any inaccuracies observed.

## C.2 CRITIQUE ANNOTATION INTERFACE

To evaluate the quality of critiques generated by VNLI-Critique and other baseline models (as described in Section 4.3), a different interface was employed. This interface presented human annotators with the original image, the factually incorrect sentence that was critiqued, and the critique generated by the model under evaluation. Figure 4 shows an example of this interface. Annotators were tasked with judging whether the provided critique accurately and relevantly identified the factual error(s) present in the original sentence when compared against the visual evidence in the image.

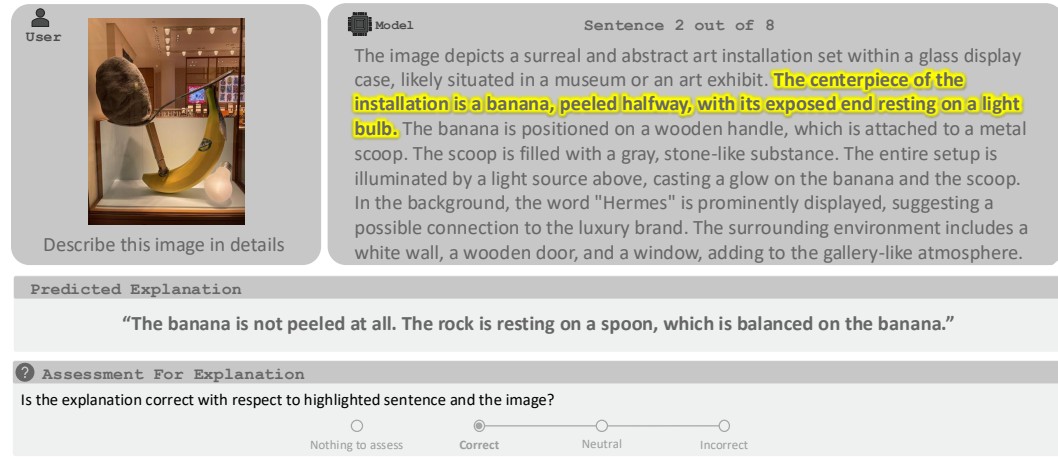

Figure 4: Example of the Critique Annotation Interface. Annotators assess if the 'Predicted Explanation' correctly identifies the error in the VLM's highlighted sentence relative to the image.

## C.3 ERROR CATEGORIZATION ANALYSIS

To better understand the common failure modes of VLMs in detailed captioning, we conducted an analysis of the error types identified by the human-provided rationales in the DOCCI-Critique benchmark. We established a set of error categories from recurring patterns in the rationales and then used an LLM to systematically classify each explanation. Table 10 presents the distribution of these error categories, offering insights into the challenges of detailed captioning.

Table 10: Breakdown of error categories in DOCCI-Critique, based on human rationales

| Error Category | Description | Percentage |
|---|---|---|
| Object Presence & Existence | The description includes objects that are not in the image or omits objects that are. | 22.99% |
| Spatial Relationship Error | An object's position or orientation is described incorrectly (e.g., "left" instead of "right"). | 16.34% |
| Attribute Error: Color & Appearance | An object's visual properties like color, texture, or shape are wrong. | 15.8% |
| Unverifiable Detail | A claim is made about something that is impossible to see or confirm from the image. | 13.25% |
| Incorrect Object Identification | An object is misidentified as something else (e.g., a toy is called a real animal). | 9.34% |
| Action & State Error | The action or state of a subject is wrong (e.g., "sitting" instead of "standing"). | 6.82% |
| Attribute Error: Count & Quantity | The number of objects described is incorrect. | 6.57% |
| Subjective or Unsupported Inference | The description includes a non-factual judgment, mood, or intent. ] | 4.89% |
| Attribute Error: Text & Numerals | Text or numbers within the image are misread. | 3.37% |
| Other | This category covers miscellaneous errors not fitting the above definitions. | 0.63% |

## D ABLATION STUDIES

We include two ablation studies to further validate our methodological choices. The first study compares our two-step Critic-and-Revise pipeline against a unified end-to-end correction model. The second study investigates the pipeline's dependency on large LLMs by testing smaller, open-source models for the revision task.

## D.1 END-TO-END VS. TWO-STEP PIPELINE COMPARISON

To validate our modular, two-step pipeline design, we compared it against a unified end-to-end (E2E) correction model. We used Gemini-2.0-Flash as the base model for both approaches to ensure a fair comparison. The E2E model was prompted to directly output a corrected sentence (or "YES" if the sentence was already correct) using the following prompt:

*"Your task is to fix the target sentence if required to. If the target text is correct, reply only "YES". If the target text does not align with the image, fix it to be aligned. Given the image and the prompt prefix <PREFIX>Claim-Prefix</PREFIX>, does the following text align with the image: <TARGET>Target-Claim</TARGET>? Remember, answer ONLY, with YES or the fixed sentence."*

The E2E setup yielded a Macro-F1 score of 0.54, only marginally better than a random classifier (0.45). This represents a substantial performance drop compared to the 0.74 Macro-F1 our two-step classification approach achieved with the same model (as shown in Table 4). This experiment confirms that simultaneously detecting and fixing sentences is a significantly more challenging task. Our two-step pipeline, by decoupling these actions, achieves higher accuracy and interpretability.

## D.2 REVISION MODEL ACCESSIBILITY AND COST

To analyze the pipeline's dependency on large proprietary models, we conducted an ablation study on the revision step using smaller, open-source LLMs. We tested Gemma3-4B and Llama3.1-8B against the proprietary Gemini-2.0-Flash model. For a fair comparison, each LLM was provided the *exact same critique* from VNLI-Critique for a sample of 1,000 sentences. The factuality of each resulting revision was then assessed using our human annotation pipeline.

The results, shown in Table 11, demonstrate that smaller, open-source models like Llama3.1-8B perform comparably to the proprietary Gemini model. This confirms that our Critic-and-Revise pipeline is not dependent on large, costly models to be effective.

Table 11: Ablation study on revision LLM. Factual accuracy of revised sentences from a 1,000-sentence sample, using identical critiques from VNLI-Critique. Note: The Gemini score differs slightly from Table 6, which used a different sample size.

| Revision LLM | Factual Revised Sentences |
|---|---|
| Gemma3-4B | 53.55% |
| Llama3.1-8B | 59.39% |
| Gemini-2.0-Flash | 61.93% |

# E QUALITATIVE EXAMPLES

To further illustrate the core components and outputs of our work, this section provides additional qualitative examples, complementing the discussions and aggregated results presented in the main paper.

Table 12 showcases another detailed entry from the DOCCI-Critique benchmark. This example highlights the fine-grained nature of our sentence-level annotations, including the multi-rater judgments on whether a sentence makes a claim about the image, its factual correctness against the visual evidence, and the diverse human-written rationales provided by annotators for any identified inaccuracies. Such examples underscore the richness of the benchmark for evaluating nuanced understanding and error analysis.

Furthermore, Table 13 provides a step-by-step walkthrough of our Critic-and-Revise pipeline operating on an image description sourced from the PixelProse Singla et al. (2024) dataset. The example demonstrates: (1) the original VLM-generated description containing factual errors, (2) the specific unfactual sentences detected by VNLI-Critique, (3) the corresponding critiques generated by VNLI-Critique, (4) the individual sentence revisions made by the LLM based on these critiques, and (5) the final, more factually accurate revised description. This illustrates the practical application of our pipeline in automatically correcting errors in detailed image captions.

Table 12: Additional DOCCI-Critique benchmark annotation example (5 raters per assessment). Details sentence-level claims, factuality, and diverse human rationales for errors, showing varied perspectives.

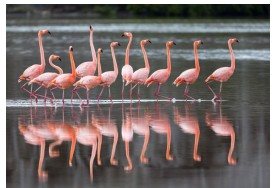

| Image | | | |
|---|---|---|---|
| **Description Sentence** | "... Looking closely, we can see eight flamingos lined up. ..." | "... They are standing in a body of water, their reflection is seen in the water, and there are trees in the background. ..." | "... Flamingos primarily eat brine shrimp, blue-green algae, small insects, mollusks, and crustaceans ..." |
| **Does the sentence include a claim about the image? (Answers from 5 raters)** | ✓, ✓, ✓, ✓, ✓ | ✓, ✓, ✓, ✓, ✓ | ✗, ✗, ✗, ✗, ✗ |
| **Is the sentence factual? (Answers from 5 raters)** | ✗, ✗, ✗, ✗, ✗ | ✗, ✓, ✓, ✗, ✓ | ✓, ✓, ✓, ✓, ✓ |
| **Rationales** | • The count of eight flamingos is incorrect; I can see at least ten.
• There are 11 flamingos in the image.
• Incorrect number of flamingos stated, there appear to be more.
• I see eleven flamingos, not eight.
• Eleven flamingos are lined up, not eight. | • I don't see any prominent trees in the background, mostly just distant, blurry foliage or land.
• The background appears to be more of a distant shoreline or low vegetation, not distinct trees. | - |

Table 13: Table 11: Step-by-step illustration of the **Critic-and-Revise** pipeline in action on a sample from the PixelProse dataset. The 'Original Description' contains several inaccuracies. 'Detected Unfactual Sentences by VNLI-Critique' highlights these errors (e.g., regarding hand position, light source, text location). 'Predicted Critiques by VNLI-Critique' provides explanations for these errors. The 'Critic-and-Revise output' shows the LLM's corrected individual sentences, guided by the critiques. Finally, the 'Revised Description' integrates these corrections into a more factually grounded paragraph.

| | |
|---|---|
| **Image** |  |
| **Original Description** | A young man with short brown hair and dark brown eyes. He is wearing a black jacket and a white shirt. He has a serious expression on his face. He is looking at the viewer with his left hand on his chin and the other holding his jacket. There is a dark background with some light coming from the left side of the image. There is text at the top of the image that says "The right to use my friends as a weapon, that is the sinful crown I shall adorn - Shu Ouma". The text is in a white font. The image is in an anime style. |

| **Detected Unfactual Sentences by *VNLI-Critique*** | He is looking at the viewer with his left hand on his chin and the other holding his jacket. | There is a dark background with some light coming from the left side of the image. | There is text at the top of the image that says "The right to use my friends as a weapon, that is the sinful crown I shall adorn - Shu Ouma". |
|---|---|---|---|
| **Predicted Critiques by *VNLI-Critique*** | He is looking at the viewer but his hands are not visible | The light is coming from the right side of the image, not the left. | The text is at the bottom of the image and not the top. |
| **Critic-and-Revise output** | He is looking at the viewer. | There is a dark background with some light coming from the right side of the image. | There is text at the bottom of the image that says "The right to use my friends as a weapon, that is the sinful crown I shall adorn - Shu Ouma". |

| | |
|---|---|
| **Revised Description** | A young man with short brown hair and dark brown eyes. He is wearing a black jacket and a white shirt. He has a serious expression on his face. He is looking at the viewer. There is a dark background with some light coming from the right side of the image. There is text at the bottom of the image that says "The right to use my friends as a weapon, that is the sinful crown I shall adorn - Shu Ouma". The text is in a white font. The image is in an anime style. |