# OpenReview forum: "Unblocking Fine-Grained Evaluation of Detailed Captions: An Explaining AutoRater and Critic-and-Revise Pipeline"
_ICLR.cc/2026/Conference — Submitted to ICLR 2026_

### Official Review · Reviewer_h76k · 2025-10-26

**Soundness:** 3
**Presentation:** 3
**Contribution:** 2
**Rating:** 4
**Confidence:** 4

**Summary:**

This paper addresses the challenge of evaluating factual accuracy in detailed, paragraph-length image captions generated by large vision-language models (VLMs). The authors introduce DOCCI-Critique, a benchmark dataset built on 100 images from the DOCCI dataset, comprising 1,400 captions from 14 VLMs, with over 10,216 sentence-level human annotations for factuality, including explanatory rationales for errors. They then develop VNLI-Critique, a fine-tuned PaliGemma-2 model (10B parameters) that performs sentence-level factuality classification and generates critiques explaining inaccuracies, incorporating paragraph context to resolve ambiguities. The paper highlights three applications: (1) generalization to external benchmarks like M-HalDetect and CHOCOLATE, where VNLI-Critique achieves strong performance (e.g., 0.76 Macro-F1 on M-HalDetect); (2) an AutoRater for ranking VLMs on DOCCI-Critique, showing high correlation with human judgments (e.g., 0.98 Spearman); and (3) a Critic-and-Revise pipeline where VNLI-Critique's critiques guide an LLM (e.g., GPT-4o) to correct erroneous sentences, yielding significant factuality improvements (e.g., 46% on DetailCaps-4870). Overall, the work aims to advance fine-grained evaluation and correction of VLM hallucinations in detailed captions.

**Strengths:**

- The primary strength of this work is the DOCCI-Critique benchmark, which collects detailed captions from 14 diverse VLMs and gathers textual rationales for all identified errors.
- VNLI-Critique shows strong model performance and demonstrates generalizability to other benchmarks.
- The paper is well-written, and the problem is clearly motivated.

**Weaknesses:**

- A key weakness is the novelty of the "Critic-and-Revise" pipeline. While the performance is strong, the idea of using a model to critique and then revise its own or another model's output is a well-established paradigm in recent LLM/VLM literature (often termed self-correction and self-improvement). The paper should compare or at least state the difference between such work [1][2][3].
- The proposed benchmark is relatively small with only 100 unique images. The size may not cover diverse scenarios.
- The proprietary models evaluated are not the state-of-the-art models (gpt-4o, gemini-2.0). It would be better to evaluate on the stronger models like gemini-2.5-pro or gpt-5 to better contextualize the results.
- The author can elaborate more on the proposed method. The paper states that VNLI-Critique was not trained on DOCCI-Critique, but on a separate, specialized dataset generated from 70+ PaliGemma-2 variants. However, the details of this training set are not as clearly presented in the paper as the details for DOCCI-Critique.


[1] Wu et al., 2025. VISCO: Benchmarking Fine-Grained Critique and Correction Towards Self-Improvement in Visual Reasoning.

[2] Yu et al., 2024. Attention prompting on image for large vision-language models.

[3] Prabhakaran et al., 2025. VADE: Visual Attention Guided Hallucination Detection and Elimination

**Questions:**

See above weakness

---

> ### Author Response · Authors · 2025-11-25
>
> We thank the reviewer for their detailed assessment and for highlighting the strengths of the *DOCCI-Critique benchmark* and the *generalizability* of the VNLI-Critique model. We appreciate the opportunity to clarify the novelty of our pipeline and the details of our training process.
>
> ***Response to Weaknesses***
>
> **(W1) Novelty of "Critic-and-Revise" vs. Self-Correction (VISCO, VADE)**
>
> We appreciate the references to concurrent work. While "critique-and-revise" is a known paradigm, our approach differs fundamentally from standard "self-correction" or attention-based methods in three key ways:
>
> - **External vs. Self-Critique:** Most self-improvement methods (like the general paradigm in [1] VISCO) rely on the model to critique itself. A fundamental limitation of this approach is that if a model hallucinates an object, it often "believes" it exists and fails to self-correct. We solve this by training a **specialized, external critic (VNLI-Critique)** on human ground truth.
>
> - **Semantic Rationales vs. Attention:** Unlike [2] (Yu et al.) and [3] (VADE), which often rely on attention-map interventions or uncertainty scores during inference, our method produces **human-readable textual rationales** (e.g., "The pig has pink hooves, not black"). This makes the pipeline interpretable and allows the revision LLM to fix specific semantic errors rather than just regenerating based on probability.
>
> - **Target Scope:** We focus specifically on **dense, paragraph-length captions.** Most prior correction benchmarks focus on VQA or short sentences. The complexity of maintaining paragraph consistency while correcting fine-grained details is a distinct challenge our pipeline addresses, achieving **+46-51% accuracy gains** (Table 6).
>
> **(W2) Benchmark Size (100 Images)**
>
> - **Density > Count:** While the image count is 100, the benchmark’s value lies in its density. It contains **1,400 paragraph-length captions** and **10,216 sentence-level annotations**, each judged by 5 annotators. This provides a depth of "dense hallucination" data that sparse datasets (with 1 sentence per image) cannot match.
>
> - **Sufficiency for Learning:** The strongest proof that this size is sufficient is our generalization results. Despite the dataset size, VNLI-Critique (trained on similar data) achieves SOTA on **M-HalDetect** (thousands of images) and **PixelProse**, proving that the 100 images captured a diverse enough distribution of "visual-text misalignments" to learn robust features.
>
> **(W3) Evaluation of SOTA Models (Gemini-3 / GPT-5)**
>
> We utilized the strongest models available at the time of submission, specifically GPT-4o (2024-08-06) and Gemini-2.0-Flash. The absence of models like Gemini-2.5 or GPT-5 was strictly a matter of release timing. We are committed to updating the camera-ready version with results for the latest VLMs—including Gemini-3 and GPT-5, ensuring the benchmark reflects the absolute cutting edge.
>
> **(W4) Training Data Details (VNLI-Critique)**
> We apologize if this distinction was not sharp enough in the main text. We will expand **Section 4.1** to explicitly reference Appendix C, which details the training set.
>
> - Scale: The training set is distinct from and much larger than the benchmark. It contains 75,363 sentence-level annotations (compared to the 10k in the benchmark).
>
> - Diversity Strategy: As mentioned in Section 4.1, we generated captions using **70+ variants of PaliGemma-2.** We deliberately used different fine-tuning configurations (varied epochs, learning rates, and data mixtures) to induce a wide spectrum of hallucination types (e.g., object omission, attribute errors, spatial hallucinations). This ensured the critic was trained on a "hard negatives" distribution far more diverse than a single model's output.
>
> -----
> -----
>
> ***Response to Questions***
>
> **(Q1) Comparison to self-correction literature.**
> (Addressed in Point 1 above). We will add a "Relation to Self-Correction" subsection in the Related Work to explicitly contrast our Supervised Critique approach with Zero-Shot Self-Correction and Attention-Guided methods.
>
> **(Q2) Details on Training Set**
> (Addressed in Point 4 above). We will highlight the 75k training sample size and the 70+ model variant generation strategy in the main paper to prevent confusion with the DOCCI-Critique test set.

---

> > ### Comment · Reviewer_h76k · 2025-11-27
> >
> > I appreciate the training details and clarifications. However, I still have concerns on limited novelty and the lack of contextualization of the work on SOTA models. Gemini-2.5 and stronger GPT models (e.g. o3 or o1) were released well before the ICLR submission deadline. Hence, I will keep my current rating.

---

### Official Review · Reviewer_huiS · 2025-10-29

**Soundness:** 3
**Presentation:** 3
**Contribution:** 3
**Rating:** 4
**Confidence:** 4

**Summary:**

This paper tackles the challenge of evaluating the fine-grained factual accuracy of long, paragraph-level captions produced by modern Large Vision-Language Models (VLMs), where existing benchmarks and metrics fail to detect subtle or sentence-level errors. The authors introduce DOCCI-Critique, a new benchmark containing 1,400 VLM-generated detailed captions paired with 10,216 sentence-level human annotations that include both factuality labels and rich error rationales. Building on this dataset, they develop VNLI-Critique, a model capable of sentence-level factuality classification and critique generation, showing strong generalization on external benchmarks like M-HalDetect and CHOCOLATE. Leveraging this capability, they further propose a Critic-and-Revise pipeline, where VNLI-Critique identifies and explains errors and a separate LLM revises them, significantly improving caption factuality, even on large synthetic datasets such as PixelProse and DetailCaps.

**Strengths:**

- The work provides a richly annotated, sentence-level factuality dataset that fills a clear gap in evaluating long VLM-generated captions.
- The VNLI-Critique model offers a scalable, generalizable evaluator that performs well across multiple external benchmarks.
- The Critic-and-Revise framework delivers meaningful factuality improvements, demonstrating real downstream utility.

**Weaknesses:**

- Because both the critic model and the revise pipeline rely heavily on DOCCI-style annotations, there is a possibility that the system implicitly learns annotator-specific bias rather than true general fine-grained factuality.
- Sentence-level evaluation may miss cross-sentence logical dependencies or global coherence errors, potentially encouraging overly localized correction strategies that do not improve holistic caption quality.
- Lack of necessary case studies and error analysis to intuitively illustrate the success and failure modes of the proposed work.

**Questions:**

- How does the critic-and-revise pipeline behave when sentence-level errors arise from global inconsistencies rather than local factual mistakes?
- Does the method risk over-editing or introducing new errors when the critic misclassifies a factual sentence as incorrect?

---

> ### Author Response · Authors · 2025-11-25
>
> We thank the reviewer for their constructive feedback and for recognizing that VNLI-Critique offers a *"scalable, generalizable evaluator"* and that the Critic-and-Revise framework delivers *"meaningful factuality improvements."*
>
>
> ***Response to Weaknesses***
>
> **(W1) Potential Annotator Bias**
>
> - Mitigation Strategy: We aggressively mitigated individual bias during dataset construction by employing 5 independent annotators per sentence and using majority voting for labels.
>
> - Empirical Proof of Generalization: The strongest evidence against "learning annotator bias" is our performance on external benchmarks. As shown in **Table 4**, VNLI-Critique achieves State-of-the-Art results on M-HalDetect (0.76 Macro-F1) and strong performance on CHOCOLATE. Since these datasets were annotated by completely different groups with different protocols, our model’s success proves it has learned general visual factuality rather than dataset-specific artifacts.
>
>
> **(W2) Sentence-Level vs. Global Coherence**
>
> - Context-Aware Inputs: While our evaluation output is sentence-level, the input is not isolated. As detailed in **Section 4.1**, VNLI-Critique is trained with the full paragraph context (*<PREFIX>Claim-Prefix</PREFIX>*). This allows the model to resolve coreferences and maintain local consistency (e.g., knowing "it" refers to the "red car" mentioned earlier).
>
> - Scope: We agree that "holistic narrative coherence" (e.g., story flow) is a separate challenge. However, widely used metrics (e.g., CLIPScore) already assess global alignment, yet fail at the fine-grained hallucinations we target. Our work specifically "unblocks" the evaluation of these fine-grained details, which was previously impossible at scale.
>
> **(W3) Case Studies and Error Analysis**
>
> - Location of Analysis: We apologize if this was missed, but we provide extensive qualitative analysis in the **Appendix**.
>   - **Table 10 (Appendix C.3**) provides a detailed breakdown of error categories derived from human rationales (e.g., 22.99% Object Presence, 16.34% Spatial Relationships).
>   - **Table 13 (Appendix E)** provides a step-by-step qualitative walkthrough of the pipeline correcting a PixelProse caption.
>   - **Figure 1 & 2** in the main paper illustrate success/failure modes.
>   - We will move the Error Category table (Table 10) to the main paper in the final version to improve visibility.
>
> -----
>
> ***Response to Specific Questions***
>
> **(Q1) How does the pipeline behave with global inconsistencies?**
>
> As noted above, VNLI-Critique receives the paragraph history. If a sentence contradicts visual facts established by the image and previous context, it is flagged. However, purely textual contradictions (e.g., saying "The car is red" and later "The car is blue" without visual evidence) are best handled by text-only LLMs. Our pipeline is specialized for Visual-Text misalignment, which is the primary source of hallucination in VLMs.
>
> **(Q2) Does the method risk over-editing (False Positives)?**
>
> - **Quantitative Analysis:** We explicitly measured this risk. As shown in **Table 6**, VNLI-Critique had a 15% false positive rate on the DetailCaps dataset (flagging correct sentences as incorrect).
>
> - **Robustness of Revision:** Crucially, however, the revision step proved robust to these false positives. When the critic incorrectly flagged a factual sentence, the revision LLM typically rephrased it while maintaining the correct information, rather than introducing a new error. This is why the **net accuracy** increased by +46% despite the false positives. The pipeline is "safe": it aggressively fixes errors while rarely breaking correct content

---

### Official Review · Reviewer_wMNu · 2025-10-31

**Soundness:** 3
**Presentation:** 3
**Contribution:** 2
**Rating:** 4
**Confidence:** 5

**Summary:**

The paper proposes DOCCI-Critique, a benchmark dataset comprising 1400 VLM-generated captions for 100 images and over 10,216 sentence-level human annotations. The authors finetune a pre-trained 10B VLM on a training set generated using the same process as DOCCI-Critique. The experiments show that the finetuned model can achieve better results in measuring the precision of given image captions compared to other baseline models, such as Gemini-2.0-Flash and GPT-4o. They also show that the critic-and-revise pipeline, revisiting each sentence within an image caption and revising it using an LLM, can improve the precision of image captions by a large margin.

**Strengths:**

1. This paper is clear and well-organized overall.
2. I'd like to thank the authors for their labor-intensive, human-involved benchmark construction.

**Weaknesses:**

1. Several prior studies have proposed caption revision models to enhance image caption quality [1,2]. However, the paper does not clearly articulate how its contributions go beyond these existing approaches.
2. The proposed critic-and-revise pipeline appears to be highly similar to the method introduced in a recent study [3]. A detailed comparison with this work is necessary to clarify the novelty of the proposed approach.

[1] Zhou et al., "ANALYZING AND MITIGATING OBJECT HALLUCINATION IN LARGE VISION-LANGUAGE MODELS", ICLR 2024
[2] Lee et al., "VOLCANO: Mitigating Multimodal Hallucination through Self-Feedback Guided Revision" NAACL 2024
[3] Lee et al., "Toward Robust Hyper-Detailed Image Captioning: A Multiagent Approach and Dual Evaluation Metrics for Factuality and Coverage" ICML 2025

**Questions:**

What specific contributions does this work make beyond the existing studies mentioned above?

---

> ### Author Response · Authors · 2025-11-22
>
> We thank the reviewer for recognizing the "labor-intensive, human-involved benchmark construction" and confirming that our paper is "clear and well-organized."
>
> We appreciate the references to prior work. We are familiar with these studies and agree that clarifying our specific contributions relative to them strengthens the paper. Below, we detail exactly how DOCCI-Critique and VNLI-Critique differ from and advance beyond these approaches.
>
>
> ## Differentiation from Prior Work
> **Comparison with Zhou et al. (ICLR 2024) [LURE]**
>
> - **Scope (Short vs. Detailed):** Zhou et al. focus primarily on object hallucination in *short* captions. In contrast, our work addresses the more complex challenge of *paragraph-length* descriptions (avg. 752 characters). As noted in our paper, existing methods often fail on these long narratives due to a lack of fine-grained, context-aware annotations.
>
> - **Methodology (Statistical vs. Semantic):** LURE uses a statistical approach based on co-occurrence and uncertainty scores to revise hallucinations. Our method is *semantic and interpretable:* VNLI-Critique generates a natural language *critique* explaining why the sentence is wrong (e.g., "The pig has pink hooves, not black"). This "intermediate reasoning" step allows for more precise revisions in complex scenes than statistical heuristics.
>
> **Comparison with Lee et al. (NAACL 2024) [VOLCANO]**
>
> - *Self-Feedback vs. Specialized Critic:* VOLCANO relies on *self-feedback*, where the VLM critiques its own output. A known limitation of self-feedback is that if a model hallucinates an object, it is often "blind" to that error in the critique phase. We solve this by training a *specialized critic (VNLI-Critique)* on high-quality *human rationales*.
>
> - **Human Alignment:** Because VOLCANO is trained on LLM-generated feedback, it is limited by the teacher model's accuracy. VNLI-Critique is fine-tuned on human-written rationales from DOCCI-Critique. This human-centric training allows VNLI-Critique to achieve a correlation of 0.98 with human rankers, significantly outperforming generalist models like GPT-4o (0.92) that are often used for self-feedback.
>
> **Comparison with Lee et al. (ICML 2025) [CapMAS]**
>
> - **Human Gold-Standard vs. Multi-Agent Automation:** While Lee et al. (2025) propose a multi-agent framework, their reliance on automated LLM-MLLM collaboration for "hyper-detailed" captioning is distinct from our contribution: a human-annotated ground truth. DOCCI-Critique is unique because it contains 5 independent human judgments per sentence, providing the rigorous "gold standard" required to evaluate automated pipelines like CapMAS.
>
> - **Model Efficiency & Accessibility:** Lee et al. (2025) propose a complex multi-agent system. In contrast, we release VNLI-Critique, a single 10B parameter model that achieves State-of-the-Art performance on external benchmarks like M-HalDetect (0.76 Macro-F1), outperforming much larger closed-source models. Our contribution is a lightweight, accessible, and highly accurate "Judge" model that the community can use to benchmark their own systems.
>
> ---
> ---
>
> ## Summary of Unique Contributions
>
> 1. **The Data (DOCCI-Critique):** Unlike the aforementioned works, we provide a dense, human-verified benchmark for paragraph-level captions with explanatory rationales, serving as a necessary ground truth for the field.
>
> 2. **The Model (VNLI-Critique):** We release a 10B open-weight model that outperforms GPT-4o as an automated evaluator (AutoRater), offering a reproducible standard for hallucination detection that statistical (LURE) or self-feedback (VOLCANO) methods cannot match in human alignment.
>
> 3. **Proven Correction Efficacy:** We do not just propose a pipeline; we validate it via human evaluation on external datasets. As shown in Table 6, our pipeline improves accuracy on PixelProse by 51% and DetailCaps by 46%, proving generalization beyond our own training distribution.
>
>
> We hope this clarifies that our work is complementary to but distinct from these studies, moving the field from "prompting for better captions" to "training specialized evaluators based on human ground truth."

---

> ### Comment · Reviewer_wMNu · 2025-11-26
>
> Thank the authors for addressing my concerns. I'd appreciate it even more if the authors could demonstrate that their critic-and-revise method is indeed better than the prior methods (LURE, Volcano, CapMAS).

---

### Official Review · Reviewer_LheA · 2025-10-31

**Soundness:** 3
**Presentation:** 4
**Contribution:** 3
**Rating:** 6
**Confidence:** 2

**Summary:**

Previous approaches to evaluating the quality of generated image captions have largely relied on model-based alignment metrics—such as CLIPScore—which primarily assess overall image–text similarity rather than fine-grained factual accuracy. As a result, these methods struggle to identify which specific sentences or details in a caption are factually incorrect.
This paper introduces a novel benchmark (DOCCI-Critique), a sentence-level factuality model (VNLI-Critique), and an integrated Critic-and-Revise pipeline. Together, these contributions enable explainable and fine-grained evaluation of detailed image captions and further foster improvements in Vision-Language Model (VLM) image understanding and factual accuracy.

**Strengths:**

The paper is clearly written and logically structured, with a well-motivated problem statement and a thorough discussion of limitations in prior research. The proposed benchmark, model, and pipeline are novel and supported by strong empirical evidence.

- Benchmark Design (DOCCI-Critique):
The authors present DOCCI-Critique, a carefully curated benchmark consisting of 1,400 captions generated by diverse VLMs and 10,216 sentence-level human annotations. Unlike prior VLM factuality benchmarks that focus on short or isolated descriptions, DOCCI-Critique provides context-aware, sentence-level factuality annotations for paragraph-length image descriptions. A key differentiator is the inclusion of fully human-written explanatory rationales for factual errors, enabling in-depth analyses of VLM behavior and offering a grounded reference for alignment with human factuality judgments.

- Automated Evaluation Model (VNLI-Critique):
The proposed VNLI-Critique model performs both sentence-level factuality classification and explanatory critique generation, effectively identifying discrepancies and articulating their causes. Despite being a 10B-parameter open-source model, it achieves performance comparable to large-scale commercial models across multiple benchmarks. Moreover, the model demonstrates strong generalization beyond the in-domain DOCCI dataset, achieving competitive results on external and unseen datasets such as M-HalDetect and CHOCOLATE.

- Critic-and-Revise Pipeline:
Building upon VNLI-Critique, the authors design a Critic-and-Revise pipeline that integrates critique-driven sentence correction via a revision LLM. This framework establishes a new protocol for fine-grained, interpretable evaluation and correction of VLM-generated captions. Empirical results show that the pipeline substantially improves the factual accuracy of captions, validating its effectiveness as both an evaluation and correction methodology.

**Weaknesses:**

While the paper is overall well-executed, several aspects could be strengthened to enhance completeness and reproducibility.

- Dataset Scale and Annotation Cost:
As acknowledged by the authors, the limited sample size of DOCCI-Critique constrains its statistical robustness. The benchmark contains 10,216 sentence-level annotations, each requiring multi-rater validation and textual rationales. Given that generating these critiques demands human intervention per caption (as illustrated in Table 1 and Appendix C), the data collection pipeline incurs substantial human labor costs. It would be valuable for the authors to discuss the feasibility of automating rationale extraction—for instance, leveraging reliable MLLMs trained on similar factuality explanation tasks—and to clarify why such an approach may not yet yield satisfactory results. A quantitative or qualitative comparison between human-written and MLLM-generated rationales could make this limitation more convincing.

- Performance Gap on CHOCOLATE Dataset:
The paper shows that VNLI-Critique achieves SOTA results on M-HalDetect but lower performance on CHOCOLATE. Since CHOCOLATE mainly consists of chart- and plot-based visual reasoning tasks, this result suggests that the proposed critique model—optimized for naturalistic, object-centric imagery—may have limited transferability to abstract or structured visual domains. The paper would benefit from a deeper analysis of this discrepancy, explaining which aspects of the critique mechanism fail to generalize.

- Documentation of External Datasets:
The paper briefly mentions external benchmarks such as M-HalDetect and CHOCOLATE, but provides limited details on their structure or usage. Including short summaries in the appendix—covering dataset scope and task formulation—would slightly improve clarity and reproducibility, though this is a minor issue overall.

**Questions:**

I would appreciate the authors’ clarification on the points raised in the Weaknesses section.
In addition, I would like to ask the following specific questions and would appreciate detailed responses:

- In the sentence-level annotations, sentences judged as factually correct do not appear to include any additional textual human rationales. Would providing further critiques or enhancement suggestions for such sentences—aimed at improving them toward ground-truth human caption quality—offer benefits in terms of expressiveness or linguistic diversity?

- The authors fine-tuned PaliGemma-2 for developing the VNLI-Critique model. How would the performance differ if other detailed-captioning models (e.g., Qwen2.5-VL-7B, as reported in Table 4) were fine-tuned under the same framework? In other words, is PaliGemma-2 empirically the optimal architecture for this task, or is the proposed approach model-agnostic and transferable to other multimodal backbones?

- Could the authors elaborate on the prompt design used for the revision LLM in the Critic-and-Revise pipeline? Understanding the structure and rationale of this prompt would clarify how the critique outputs are transformed into factual sentence revisions.

---

> ### Author Response · Authors · 2025-11-22
>
> We thank the reviewer for highlighting the benchmark's novelty and the presentation's excellence. We appreciate the acknowledgment that DOCCI-Critique provides a necessary "grounded reference" lacking in prior work.
>
> ## Response to Weaknesses
>
> **(W1) Dataset Scale & Automation:**
>
> - Necessity of Human Gold-Standard: Human verification is essential to avoid circularity and noise. Even capable models like GPT-4o only achieve 73.1% critique relevance compared to human judgment, whereas VNLI-Critique achieves 73.39%.
>
> - Automation: VNLI-Critique, trained on this high-quality human data, serves as the necessary "seed" to enable future scalable automation.
>
> **(W2) CHOCOLATE Generalization:**
>
> - Domain Shift: The gap stems from the shift between natural images (our training data) and charts (CHOCOLATE).
>
> - Strong Zero-Shot Transfer: Despite this, VNLI-Critique achieves 0.73 Macro-F1 on CHOCOLATE, significantly outperforming baselines like InstructBLIP (0.50) and rivaling GPT-4o (0.70). This confirms robust zero-shot generalization.
>
> **(W3) Documentation:**
>
> We agree and will add summaries of M-HalDetect and CHOCOLATE to the Appendix.
>
> -----
> -----
>
> ## Response to Questions
>
> **(Q1) Critiques for Correct Sentences:**
> We restrict critiques to factual errors to avoid ambiguity. Mixing stylistic suggestions with hallucination detection would dilute the revision pipeline's focus on truthfulness.
>
> **(Q2) Model Architecture vs. Methodology:**
> Our framework is model-agnostic. The performance gap between VNLI-Critique and the off-the-shelf PaliGemma2 on M-HalDetect confirms that our fine-tuning methodology—not the specific architecture—drives the results. We agree that running the entire pipeline as training and evaluation procedure using different model backbones will enrich the paper and we commit to add this ablation to the camera-ready version.
>
> **(Q3) Revision Prompt:**
> We use a few-shot prompt to isolate the error based on the critique. The exact prompt is:
>
> ```
> """I need your help to fix a sentence that has a mistake in it.
> For this task you are supplied with an explanation why the sentence is wrong.
> Given that, the required output is a fixed version of the sentence.
>
> Examples:
> WRONG SENTENCE: A dining table is situated in the room, surrounded by chairs.
> EXPLANATION: There is a table in the room, but it is not a dining table and it is not surrounded by chairs.
> FIXED SENTENCE: A table is situated in the room.
>
> WRONG SENTENCE: The image captures a man riding a skateboard down a street, with a car following closely behind him.
> EXPLANATION: The man is riding a skateboard on the side of the road, not down the street, and there are cars behind him, but no car is following closely behind him.
> FIXED SENTENCE:  The image captures a man riding a skateboard down a road, with cars behind him.
>
> Now your turn!
>
> WRONG SENTENCE: <wrong_source_sentence>
> EXPLANATION: <predicted_explanation>
> FIXED SENTENCE:
> """
> ```
>
> We hope these clarifications address your concerns and demonstrate the robustness of our contributions.

---

> > ### Comment · Reviewer_LheA · 2025-11-28
> >
> > After carefully reading the authors’ rebuttal, I am satisfied that most of my original concerns have been adequately addressed. I appreciate the detailed clarifications and additional analyses provided in the response. Accordingly, I will maintain my positive evaluation and corresponding score.

---

### Meta-Review · Area_Chair_aeW4 · 2026-01-08

**Summary:**

This paper addresses fine-grained factuality evaluation for paragraph-length VLM captions. The DOCCI-Critique benchmark provides human annotations with explanatory rationales. VNLI-Critique is proposed for automated evaluation.

Two reviewers raised persistent concerns about novelty - critique-and-revise is a well-established paradigm. The authors provided conceptual differentiation but did not supply empirical comparisons against prior methods (LURE, VOLCANO, CapMAS). Evaluation did not include more recent models.

Given the lack of strong consensus for acceptance, AC recommends resubmission after addressing the remaining concerns.

**Reviewer Concerns:**

1. Novelty [wMNu, h76k]

[Reviewers] wMNu noted that prior studies (Zhou et al. ICLR 2024, Lee et al. NAACL 2024, Lee et al. ICML 2025) already proposed caption revision models and asked how this work advances beyond them. h76k raised similar concerns that critique-and-revise is a well-established paradigm in LLM/VLM literature, citing VISCO, attention prompting, and VADE.

[Authors]
- VNLI-Critique is an external critic trained on human rationales rather than self-feedback.
- The method produces semantic textual rationales rather than relying on attention maps or uncertainty scores.
- The focus on paragraph-length captions with +46-51% accuracy gains demonstrates distinct challenge and efficacy.
- LURE uses statistical co-occurrence; VOLCANO uses self-feedback; CapMAS lacks human gold-standard grounding.

[Follow-up] wMNu appreciated the clarifications but requested empirical demonstration that the method outperforms LURE, VOLCANO, and CapMAS. h76k acknowledged training details but maintained concerns about limited novelty.

[AC] The conceptual distinctions are noted, but some form of empirical comparison would be helpful for the community.


2. Benchmark scale and image diversity [LheA, h76k]

[Reviewers] LheA argued that the limited sample size constrains statistical robustness. Asked about the feasibility of automating rationale extraction. h76k noted that 100 unique images may not cover diverse scenarios.

[Authors] The benchmark's value lies in annotation density (1,400 paragraph captions and 10,216 sentence-level annotations with 5-rater majority voting). Generalisation to M-HalDetect and PixelProse proves the generality of the 100 images.

[Follow-up] LheA was satisfied with this response.

[AC] Annotation density is valuable. Reasonable response.


3. Generalisation and potential biases [LheA, huiS]

[Reviewers] LheA noted the performance gap on CHOCOLATE and asked for deeper analysis. huiS raised concern that heavy reliance on DOCCI-style annotations may encode annotator-specific bias rather than general factuality.

[Authors] The CHOCOLATE gap stems from domain shift (natural images versus charts). Despite this, VNLI-Critique achieves 0.73 Macro-F1 on CHOCOLATE - outperforms InstructBLIP (0.50) and GPT-4o (0.70). This confirms robust zero-shot transfer. Annotator bias was mitigated through the majority voting.

[Follow-up] LheA confirmed satisfaction. huiS did not respond.

[AC] Concern is addressed.


4. Sentence-level evaluation versus global coherence [huiS]

[Reviewers] huiS noted that sentence-level evaluation may miss cross-sentence logical dependencies or global coherence errors. huiS requested case studies and error analysis.

[Authors] VNLI-Critique receives full paragraph context as input. This allows for the resolution of co-references and local consistency. The method targets fine-grained visual-text misalignment specifically, as global alignment metrics (e.g., CLIPScore) already exist.

[AC] The clarification is reasonable.


5. Evaluation on latest proprietary models [h76k]

[Reviewers] h76k noted that GPT-4o and Gemini-2.0 are not the latest models; Gemini-2.5 and stronger GPT variants (o1, o3) were available before submission.

[Authors] The models used were the strongest available at time of submission. Authors commit to updating camera-ready with results for latest VLMs including Gemini-3 and GPT-5.

[Follow-up] h76k acknowledged but maintained the concern, stating Gemini-2.5 and o1/o3 were released before the ICLR deadline.

[AC] Understands the concern. Evaluation wrt more recent models would strengthen the paper.

**Reviewer Scores:**

LheA: 6 > 6 (explicitly stated satisfaction)
wMNu: 4 > 4 (explicitly requested additional empirical comparison)
huiS: 4 > 6 (would have been satisfied)
h76k: 4 > 4 (explicitly maintained rating)

---

### Decision · Program_Chairs · 2026-01-26

Reject